# Galloping Bubbles

Jian H. Guan[1,3], Saiful I. Tamim[1,3], Connor W. Magoon [1,3], Howard A. Stone [2] & Pedro J. Sáenz [1] ✉

Despite centuries of investigation, bubbles continue to unveil intriguing dynamics relevant to a multitude of practical applications, including industrial, biological, geophysical, and medical settings. Here we introduce bubbles that spontaneously start to 'gallop' along horizontal surfaces inside a vertically-vibrated fluid chamber, self-propelled by a resonant interaction between their shape oscillation modes. These active bubbles exhibit distinct trajectory regimes, including rectilinear, orbital, and run-and-tumble motions, which can be tuned dynamically via the external forcing. Through periodic body deformations, galloping bubbles swim leveraging inertial forces rather than vortex shedding, enabling them to maneuver even when viscous traction is not viable. The galloping symmetry breaking provides a robust self-propulsion mechanism, arising in bubbles whether separated from the wall by a liquid film or directly attached to it, and is captured by a minimal oscillator model, highlighting its universality. Through proof-of-concept demonstrations, we showcase the technological potential of the galloping locomotion for applications involving bubble generation and removal, transport and sorting, navigating complex fluid networks, and surface cleaning. The rich dynamics of galloping bubbles suggest exciting opportunities in heat transfer, microfluidic transport, probing and cleaning, bubble-based computing, soft robotics, and active matter.

Bubbles often appear to have a life of their own. Da Vinci[1] was a pioneer in documenting their capricious behavior, observing how rising bubbles spontaneously abandon straight paths for mesmerizing helices—a paradox that has persisted for centuries[2,3]. When exposed to acoustic excitations, bubbles may also transition from in-place pulsations, opting instead to 'dance' along zigzag paths reminiscent of Brownian motion[4,5]. Bubbles can exhibit violent dynamics: under sudden pressure changes, they may swiftly collapse, giving rise to shock waves that pose significant harm to machinery—a phenomenon known as cavitation[6,7], which some crustaceans even leverage to stun prey[8]. In some instances, the implosion may become so intense that the bubbles emit a spark of light[9]. Bubbles may also appear to violate Archimedes' principle[10], e.g., bubbles in vertically oscillating baths may overcome buoyancy and sink contradicting common intuition[11–13]. Waves of bubbles surging downwards may be also observed in carbonated drinks[14],

resulting from convection currents in the core of the glass[15,16]. Additional examples showcasing the fascinating dynamics of bubbles can be found in innumerable settings, spanning soft and biological matter[17], geophysical flows[18], and industrial processes[19].

Here, we demonstrate that a bubble inside a vertically vibrated fluid chamber may spontaneously break symmetry and start to 'gallop' along the upper wall, self-propelled through a resonant interaction between its vibration modes (Fig. 1a, Supplementary Movie 1). By adjusting the vibrational forcing, these galloping bubbles may be tuned to transition between different domain exploration modes, including rectilinear, orbital, and run-and-tumble motion (Fig. 1b–d). Moreover, similar to jellyfish and other marine invertebrates that deform their bodies to swim without vortex shedding[20–22], we show that galloping bubbles leverage inertial fluid forces to advance, thus enabling propulsion in inviscid flows where viscous traction is not possible[23].

[1]Department of Mathematics, University of North Carolina, Chapel Hill, NC, USA. [2]Department of Mechanical & Aerospace Engineering, Princeton University, Princeton, NJ, USA. [3]These authors contributed equally: Jian H. Guan, Saiful I. Tamim, Connor W. Magoon. ✉e-mail: saenz@unc.edu

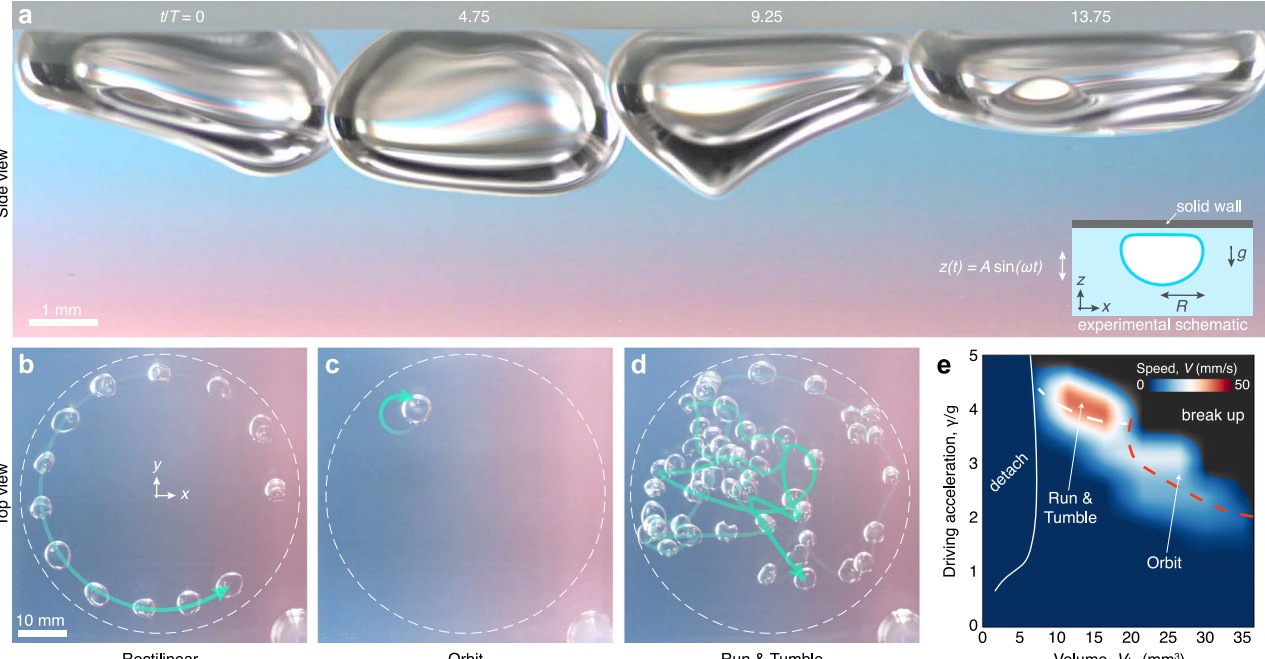

**Fig. 1 | Galloping bubbles. a** Time sequence illustrating a self-propelling bubble under the upper boundary of a vertically vibrating fluid chamber (Supplementary Movie 1), exhibiting shape oscillations reminiscent of galloping motion. The inset includes a schematic of the setup, and $T = 2\pi/\omega$ is the oscillation period. The bubbles were backlit through a gradient filter to enhance their aesthetic appeal. Different bubble sizes and vibrational forcings produce diverse domain exploration modes, observed from the top view (**b**–**d**), Supplementary Movie 2). A bubble may gallop in (**b**) steady rectilinear motion in an infinite bath, with its trajectory becoming circular here due to the chamber's boundary (dashed line, see Methods, Experiments). Time progression is indicated by the green arrow, with increasing opacity indicating later times, and the image sequence intervals are 20 $T$. Depending on the bubble volume, increasing the forcing amplitude $A$ may cause the bubble's trajectory to curve, leading to orbital states (**c**), which can develop anywhere within the chamber, or become jagged with random sharp turns (**d**) reminiscent of 'run-and-tumble' motion[36,37]. **e** A phase map at $f = 40$ Hz illustrates the dependence of the bubble's dynamics on the driving acceleration and bubble volume, which also includes detachment from the wall for smaller bubbles and breakup for large amplitudes.

Given the multifaceted physics of bubbles, harnessing their dynamics may be challenging but holds significant potential rewards. Actuated-bubble technologies have been developed for various purposes across numerous fields. In medicine, bubbles have been used for manipulating and assessing the mechanical properties of cells[24], as well as for drug and gene delivery[25]. Also, bubbles have been exploited as a cleaning tool to treat wastewater[26], prevent biofouling[27], and clear bacteria from narrow conduits[28,29]. Manipulating bubble motion is particularly valuable in boiling-based heat transfer, a common method for cooling electronic microdevices, where unremoved bubbles near the heat source can significantly reduce heat transfer efficiency[30]. This challenge is exacerbated in micro-gravity settings, such as those encountered by microchips in satellites and spacecrafts, where the absence of buoyancy complicates bubble evacuation[31–33]. To foster new bubble-based technologies, we present a series of proof-of-concept experiments that showcase the versatility of the galloping dynamics and inform new methods for producing bubbles with tunable size, sorting and removal of bubbles, navigating fluid networks, and cleaning surfaces.

## Results
### Galloping bubbles
In our experiments, we inject an air bubble of volume $V_b$ into a relatively large transparent fluid chamber filled with silicone oil with a kinematic viscosity of $\nu = 5$ cSt. Buoyancy holds the bubble against the inner surface of the chamber's top wall, where it rests on a thin fluid film due to the perfect wetting of silicone oil, rendering the bubble highly mobile (Fig. 1a). The bubble is placed far away from the chamber's vertical walls to ensure that they have no influence on the dynamics. To facilitate the excitation of vibration modes similar to

those forming the spectrum of a spherical bubble[34,35], we choose volumes that result in bubbles adopting nearly hemispherical equilibrium shapes (Supplementary Fig. S1). This configuration emerges when the bubble radius, $R$, measured with a spherical cap fitted to the underside, is comparable to the liquid's capillary length, $l_c = \sqrt{\sigma/\rho g} = 1.48$ mm, where $\sigma$ is the surface tension, $\rho$ the liquid density, and $g$ the gravitational acceleration. The dynamics of larger bubbles, which are flattened by gravity when $R \gg l_c$, and smaller bubbles, which become spherical when $R \ll l_c$, are more intricate due to the departure from spherical geometry and influence of the wall, respectively.

To activate a resonant interaction between the natural vibration modes of the bubble, we drive it out of equilibrium with an electromagnetic shaker that subjects the fluid chamber to a sinusoidal vertical motion, $z(t) = A \sin(\omega t)$, where $A$ is the maximum bath displacement, $f = \omega/2\pi = 40$ Hz the driving frequency, and $t$ time. In the frame moving with the chamber, the bubble is thus subject to an effective variable gravitational field, $\boldsymbol{G}(t) = (-g + \gamma \sin(\omega t))\hat{\boldsymbol{z}}$ with $\gamma = A\omega^2$, that excites shape oscillations. At low forcing, when the bath displacement is small relative to the characteristic bubble size, $0 < A/R \ll 1$, the bubble exhibits symmetric harmonic shape oscillations (Supplementary Fig. S2). Notably, as the driving amplitude is increased beyond a critical threshold, $A_G$, the bubble undergoes a spontaneous symmetry breaking about the vertical axis, and starts to self-propel along the fluid chamber with harmonic oscillations reminiscent of galloping motion (Fig. 1a, Supplementary Movie 1).

Galloping bubbles display various domain exploration modes (Fig. 1b–d, Supplementary Movie 2). By tuning the bubble volume and driving, the propulsion mode may transition between rectilinear motion, where bubbles travel in straight paths (Fig. 1b), orbital motion,

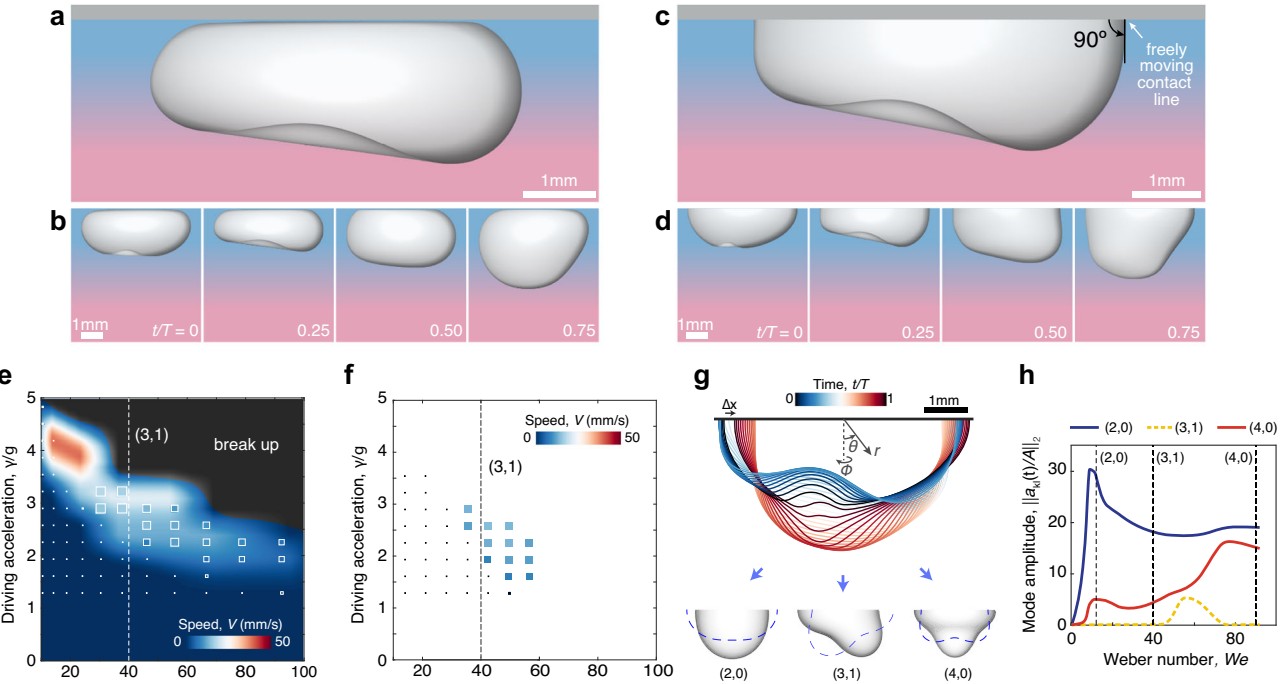

**Fig. 2 | Hemispherical galloping bubbles and their spectrum. a, b** Direct numerical simulations capture the galloping dynamics (Supplementary Movie 3). As in the experiments, where a thin fluid layer separates the bubble from the boundary, no contact line is formed at the top wall in our model. **c, d** Simulations demonstrate that the same galloping instability arises for sessile bubbles with a freely moving contact line and 90° contact angle. **e** A quantitative comparison of bubble speed between experiments (background) and simulations of full bubbles (marker colors and sizes reflect the galloping speed) across a range of normalized driving acceleration, $\gamma/g$, and dimensionless frequency given by the Weber number, *We*. Galloping motion is observed near *We* = 40, corresponding to the natural frequency of the (3,1) vibration mode (indicated by the vertical dashed line). **f** Hemispherical bubbles exhibit galloping dynamics in the same region of the phase map, and (**g**) their shape oscillations, which lead to a net displacement $\Delta x$ per period, are primarily composed of the $(k,l) = (2,0)$, (3,1), and (4,0) spherical harmonics, $Y_{kl}(\theta,\varphi)$. **h** Mode dominance vs *We* number for fixed driving $A/R = 0.08$ characterized via the $L_2$ norm of the instantaneous amplitude $\|a_{kl}(t)/A\|_2$. The emergence of the (3,1) harmonic coincides with the onset of galloping.

characterized by closed circular trajectories (Fig. 1c), and run-and-tumble motion, in which the bubble moves chaotically, mimicking the search strategy of a variety of organisms[36,37] (Fig. 1d). We perform a systematic series of experiments to identify the parameter regime where the different propulsion modes emerge (Fig. 1e), which reveals additional dynamics, including detachment from the wall for smaller bubbles and breakup at high forcings. Bubble breakup occurs as a result of large-amplitude shape oscillations, leading either to the bubble dividing into two similarly sized pieces or to the ejection of smaller bubbles due to interface pinch-off. Notably, the bubbles may achieve relatively high steady speeds. Defining the average bath speed as $V_{bath} = 4A/T$, we observe steady galloping speeds in the same order of magnitude, up to $V \sim 0.44 V_{bath}$, which highlights the capacity of the galloping instability to efficiently transform vertical into lateral motion. Moreover, the galloping bubbles exhibit swimming efficiencies comparable to those achieved through biological evolution. Using the bubble radius, $R$, as the characteristic body length and the driving frequency, $f$, as the natural beat frequency, we find that galloping bubbles can reach swimming speeds as high as $V \sim 0.5Rf$, which falls within the typical range observed in fish and cetaceans[38].

## Hemispherical bubbles

Owing to the deformation caused by buoyancy and the upper wall, the detailed geometry of galloping bubbles complicates the analysis of their spectrum and self-propulsion mechanism. Are these geometric details crucial, or may the galloping locomotion arise in other bubble configurations? To answer this question, we complement our experiments with simulations of the Navier-Stokes equations, (see Methods, Simulations). Our simulations capture the galloping dynamics of

bubbles similar to those investigated in experiments, showing nearly identical shape oscillations (Fig. 2a, b). Moreover, bubbles of various sizes exhibit galloping speeds consistent with experimental observations, as shown in Fig. 2e, where comparisons are made in terms of the normalized driving acceleration, $\gamma/g$, and the dimensionless Weber number, $We = \rho\omega^2 R^3/\sigma$. Notably, we then leverage simulations to demonstrate that the same symmetry breaking arises in hemispherical bubbles with freely moving contact lines (Fig. 2c, d, Supplementary Movie 3) across the same range of bubble volumes and driving amplitudes (Fig. 2f) indicating that, effectively, the same vibration modes are responsible for motion in both configurations.

## Vibration spectrum

Demonstrating that the galloping instability extends to nearly hemispherical bubbles not only highlights the universality of this symmetry breaking but also facilitates the rationalization of the propulsion mechanism through a spectral analysis, where the bubble shape can now be decomposed into a minimal number of modes. We thus project the bubble interface $r(\theta, \varphi, t) = R + \sum_{k,l} a_{kl}(t)Y_{kl}(\theta, \varphi)$ onto an orthogonal basis of spherical harmonics, $Y_{kl}$, each with temporal amplitude $a_{kl}(t)$, and $k + l$ = even to satisfy the 90° contact-angle condition (Fig. 2g). In the idealized case of a bubble in an inviscid flow[34,35], the resonant frequency of each mode in the linear regime is given in dimensionless form by the Rayleigh equation, $We_k = (k^2 - 1)(k + 2)$ (Methods, Theory).

The spectral analysis of hemispherical galloping bubbles reveals that their shape is dominated by two axisymmetric modes, $(k, l) = (2,0)$ and (4,0) with natural frequencies $We = 12$ and 90, respectively, along with an asymmetric mode, (3,1) with natural frequency $We = 40$, whose amplitudes increase when the driving frequency is in the vicinity of

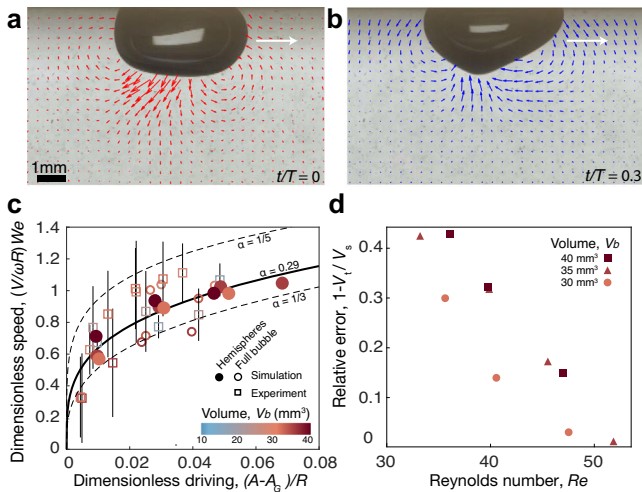

**Fig. 3 | Propulsion mechanism. a, b** Experimental visualization of flow fields around a galloping bubble reveals the interface (**a**) pushing ambient fluid towards the bubble's back when moving downwards, and (**b**) drawing fluid from its front when moving upwards (Supplementary Movie 4). **c** Rectilinear galloping bubbles obey a power scaling law for the dimensionless galloping speed $V/\omega R$ in terms of the dimensionless driving $(A-A_G)/R$ and Weber number $We$, which collapses the simulated hemispherical bubbles with a power $\alpha = 0.29$. **d** The relative deviation between the galloping speed in simulated hemispherical bubbles, $V_s$, and that expected from inviscid theory, $V_t$, decreases with the Reynolds number, indicating that galloping bubbles leverage inertial forces for propulsion.

their resonant frequencies (Fig. 2g, h). Small deviations from the natural frequencies are expected due to nonlinear deformations and the influence of viscosity (Methods, Theory). Notably, the asymmetric (3,1) mode is absent below the galloping threshold, $(A < A_G)$, and only appears in the spectrum when the bubble starts to self-propel when the driving amplitude exceeds the galloping threshold, $(A > A_G)$. Moreover, unlike Faraday waves[39,40], which arise in a stationary fluid layer (in the moving frame), the galloping motion arises from the parametric excitation of an asymmetric mode atop an oscillatory base state. If the asymmetric (3,1) mode was excited in isolation, the bubble would exhibit symmetry over a period and, consequently, no net motion. The coupling between the asymmetric mode and the oscillatory base state is thus essential to render the bubble shape temporally asymmetric, ultimately resulting in net propulsion.

**Propulsion mechanism**

When a bubble gallops, its asymmetric shape oscillations induce a net circulation of fluid: liquid is drawn from the front of the bubble when the interface moves upwards, and later is pushed backwards when the interface moves downwards (Fig. 3a, b, Supplementary Movie 4). To demonstrate that the resulting net propulsion is directly correlated to the emergence of the (3,1) mode, we seek a scaling law for the galloping speed, $V = V(\omega, R, A - A_G, We)$, in terms of the driving frequency and bubble size, which define the characteristic flow speed, $\omega R$, the distance from the galloping threshold, $A - A_G$, which serves as a proxy for the amplitude of the (3,1) mode, and the Weber number, which locates resonances through the driving frequency and bubble properties. Dimensional analysis then indicates a functional relationship $V/(\omega R) = F\left(\frac{A-A_G}{R}, We\right)$, which we observe can be simplified to $(V/\omega R)We = c\left(\frac{A-A_G}{R}\right)^{\alpha}$.

Indeed, experimental and simulated bubbles in the rectilinear regime confirm the suitability of this power-law scaling (Fig. 3c), which collapses the hemispheres onto $\alpha = 0.29$ and $c = 2.38$, with small variations for the full bubbles $(1/5 < \alpha < 1/3)$. Bubbles exhibiting orbital and run-and-tumble motion at higher driving have been excluded due to

their spectrum encompassing a broader range of shape oscillation modes coupled together.

Many aquatic organisms[20,21,41] and vehicles[22,42–45] that utilize periodic oscillations for propulsion rely on vortex shedding, which implies a dependence on viscosity that other organisms have managed to circumvent by leveraging inertial forces via body deformations[23,46,47]. To demonstrate that galloping bubbles represent a minimal realization of this distinct type of geometric swimming, we compare the galloping speed in simulations with that expected from purely inertial forces within an inviscid flow[23], which is given by $v(t) = \frac{-3}{\pi R^3}\int_S \phi(\hat{\boldsymbol{n}} \cdot \hat{\boldsymbol{t}})\,dS$. Here, $\hat{\boldsymbol{n}}$ and $\hat{\boldsymbol{t}}$ are the unit vectors normal to the free surface $S$ and tangential to the bubble trajectory, respectively, and $\phi$ is a scalar potential describing the inviscid flow, which may be approximated from the interface deformations in our simulations (Methods, Theory). The instantaneous velocity $v(t)$, which results from the balance between the liquid's added mass and inertial forces due to the interface deformations, may be integrated over an oscillation period, $T$, to obtain the steady galloping speed, $V = \frac{1}{T}\int_0^T v(t)\,dt$. We find that the relative error between the theoretical prediction, $V_t$, and the actual bubble speed in our simulations, $V_s$, decreases with increasing Reynolds number, $Re = A\omega R/\nu$, becoming $1 - V_t/V_s < 0.1$ for $Re > 45$ (Fig. 3d), which demonstrates that galloping bubbles exploit inertial forces for propulsion, eliminating any dependence on viscous effects.

**Oscillator model**

Is the galloping symmetry breaking exclusive to bubbles, or can this propulsion mechanism be realized in other systems? To shed light on this question, we derive a reduced oscillator model[48] that encapsulates the fundamental physics underlying galloping bubbles. Inspired by the classical studies of Rayleigh[34] and Lamb[35], who demonstrated that the amplitude of a bubble's vibration modes $Y_{kl}$ is governed by a mass-spring-damper model, we conceptualize a galloping bubble as a pendulum of point mass $M$, representing the displaced liquid, and equilibrium length $L$ that is parametrically excited via vertical oscillations of its pivot (Fig. 4a). The pendulum is permitted to deform as a spring with constant $k$ to account for the restorative influence of surface tension. To mimic the hydrodynamic interactions between the bubble and its enveloping liquid, the mass is subject to fluid-like inertial forces, $\propto \boldsymbol{u}^2$. After appropriate re-scaling, the dimensionless oscillator model for the position vector relative to the pivot, $\boldsymbol{r} = (r, \theta)$, becomes,

$$\ddot{\boldsymbol{r}} + p|\dot{\boldsymbol{r}}|\dot{\boldsymbol{r}} + \zeta\dot{\boldsymbol{r}} + \frac{1}{\kappa}(r-1)\hat{\boldsymbol{r}} - \boldsymbol{g}(t) = 0, \tag{1}$$

where $\boldsymbol{g}(t) = (\delta + \varepsilon\cos t)\hat{\boldsymbol{x}}$ is the effective gravity, with amplitude $\varepsilon = A/L$, $\hat{\boldsymbol{r}}$ and $\hat{\boldsymbol{x}}$ are the unit radial and vertical vectors, and $p, \zeta, \kappa, \delta$, are the proportionality coefficients corresponding to liquid inertia, viscous damping, pendulum deformability, and a steady gravitational force, respectively (Methods, Theory).

The oscillator model (1) in the weakly-deformable limit $(0 < \kappa \ll 1)$ reveals the essence of the symmetry-breaking mechanism responsible for galloping bubbles (Methods, Theory). Similar to the symmetric base state seen in the bubble dynamics, the mass initially undergoes stable vertical oscillations at low forcing (Fig. 4b). Beyond a certain driving threshold $A_G$, which may be predicted using Floquet theory (Supplementary Fig. S5), a parametric instability develops in the angular direction, inducing lateral oscillations that are saturated by the system's nonlinearities. Notably, these lateral oscillations couple with the underlying compression-extension cycles, thus spontaneously causing the mass motion to acquire angular momentum (Fig. 4c). To illustrate how this symmetry breaking may lead to self-propulsion, we consider the pendulum to be attached to a sliding pivot with position $(0, y_p(t))$, mass $M_p$, and frictional coefficient $D$ (Fig. 4d, and Methods, Theory). By solving the resulting coupled system ((25), Methods) in a regime where the pivot displacement is small relative to that of the

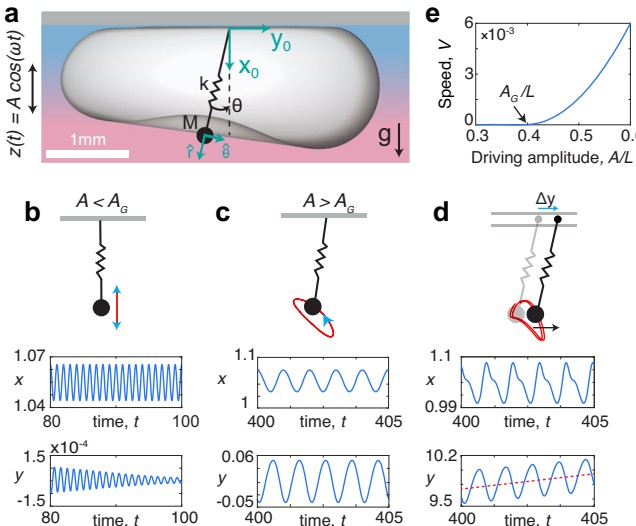

**Fig. 4 | Oscillator model. a** A weakly deformable pendulum with spring constant $k$, point mass $M$, and equilibrium length $L$, subject to vertical oscillations exhibits a symmetry breaking analogous to that of galloping bubbles. **b–d** The pendulum's trajectory (red) is depicted through the instantaneous $x_0 - y_0$ coordinates relative to the pivot. **b** At low forcing amplitude, the motion is purely vertical. **c** Above a critical threshold, $A > A_G$, the mass acquires angular momentum due to the coupling between vertical base oscillations and spontaneously emerging lateral oscillations. **d** If the pivot is allowed to slide, the mass motion gives rise to horizontal translation, where (**e**) the steady propulsion speed $V$ is proportional to $A - A_G$. The model parameters are $\kappa = 0.05$, $p = 0.3$, $\zeta = 0.02$, $\delta = 1.1$ and (**b**) $\varepsilon = 0.2$, (**c**) $\varepsilon = 0.4$, and (**d**) $\varepsilon = 0.6$, $\zeta_p = 0.2$, $\xi = 100$ (Methods, Theory).

pendulum (e.g. $M_p \gg M$), the motion of the swinging mass remains largely unaffected by the pivot's motion and generates a net horizontal force that propels the pendulum forward (Fig. 4d) at a steady speed, which increases with the distance beyond the instability threshold (Fig. 4e). No net propulsion is observed if the pendulum is rigid[49] ($\kappa = 0$), where the motionless base state prevents a symmetry breaking, or highly deformable ($\kappa \gg 1$), where oscillations become exceedingly chaotic.

### Proof-of-concept applications

To demonstrate the potential of the galloping instability to unlock new technological opportunities across various practical settings, we present a series of proof-of-concept experiments in Fig. 5 (see Supplementary Movie 5). The galloping mechanism may be leveraged to evacuate bubbles from a nucleation point, such as those encountered in boiling processes (Fig. 5a), and selectively produce bubbles of different sizes by tuning the forcing frequency, which determines the critical size at which bubbles will gallop (Fig. 5b). Galloping bubbles also exhibit affinity to follow lateral walls, suggesting new possibilities for size-dependent bubble sorting methodologies (Fig. 5c) and enabling their navigation through complex flow networks and mazes (Fig. 5d). This attraction to vertical walls is also captured in simulations, which reveal how sidewalls induce an additional symmetry breaking in the bubble shape and flows perpendicular to the galloping direction, holding the bubble against the wall (Supplementary Fig. S6). Additionally, galloping bubbles offer a non-invasive cleaning method for removal of microparticles from solid boundaries (Fig. 5e), where particles are swept downward beneath the bubble by the oscillation-induced flows (Supplementary Fig. S7). No noticeable particle depletion was observed when the fluid the chamber was vibrated without the galloping bubble, indicating that bubble-induced flows are the primary force overcoming particle adhesion.

## Discussion

Bubbles remain a hydrodynamic gift that keeps on giving, continuously drawing increasing interest due to their ubiquity and myriad of practical applications. Through experiments, simulations, and theory, we have unveiled a symmetry-breaking instability whereby vertically-vibrated bubbles spontaneously start to gallop along horizontal solid boundaries. These bubbles exhibit various strategies of exploring their surroundings, including rectilinear, orbital, and run-and-tumble motions, easily adjustable through external forcing. The galloping symmetry breaking constitutes a robust self-propulsion mechanism, observed in bubbles with and without a thin liquid film separating them from the wall. By leveraging inertial forces, galloping bubbles move without relying on shedding vortices for propulsion, thus expanding their applicability to inviscid flows. Moreover, we develop a minimal oscillator model that encapsulates the essence of the bubble self-propulsion, which suggests the feasibility of realizing the galloping locomotion in other systems. In a broader context, we present a series of proof-of-concept experiments to showcase the potential of the galloping instability for practical applications, including selectable-size generation, sorting and removal of bubbles, maneuvering through complex fluid networks and mazes, and surface cleaning tasks. Galloping bubbles thus open new technological avenues in a range of settings, including gas removal in heat transfer systems[31–33], microfluidic[50] cleaning[28], transport[25], delivery[25,51] and mechanical probing[24], bubble-based computing[52], aquatic soft robotics[22], and fluid-mediated active matter[53].

## Methods

### Experiments

Bubble chamber—The fluid chamber housing the bubble is made out of 5 mm thick clear acrylic to enable the observation of bubbles from all sides. The top wall, against which the bubbles rest, was 3D-printed using a high-precision 3D printer (Formlabs, $25\,\mu m$ layer thickness and XY resolution) to manufacture a flat surface surrounded by a gentle encircling slope which serves to redirect the bubble without influencing its speed significantly. We performed tests where the top wall was a completely planar acrylic sheet to confirm the observed galloping dynamics are the same. The chamber is filled with 5 cSt silicone oil (1 cSt = 1 mm² s⁻¹) with density $\rho = 918\,kg\,m^{-3}$ and surface tension $\sigma = 19.7\,mN\,m^{-1}$. Owing to the high wettability of silicone oil, the liquid fully wets the top boundary, ensuring the maintenance of a thin liquid film between the bubble and the top wall. A small opening between the lid and the vertical walls ensures that the reference pressure is atmospheric. The size of the chamber is very large relative to the bubble size, $L \gg R$ to ensure that the bottom and vertical walls play a negligible role in the bubble dynamics.

Bubble injection—We inject bubbles of known volumes, $V_b$, into the bubble chamber through a small opening in the top wall using a calibrated syringe pump (World Precision Instruments, AL-4000). The volume of the injected bubble is checked by means of two additional methods. Firstly, we compute the bubble volume from the mass difference between a fully-filled chamber and one with an injected bubble. Secondly, we measure the volume of the injected bubble using an in-house MATLAB algorithm which computes the bubble volume from its cross-sectional area measured from side-view images and by assuming azimuthal symmetry. We find that the three methods yield a volume within 5% error.

Levelling—The liquid film separating the bubble from the top wall renders the bubble highly mobile; any departure from the horizontal in the levelling of the top wall may thus cause the bubble to move due to buoyancy. To ensure that the motion of the galloping bubbles are not significantly influenced by any surface tilt, our vibrating set-up undergoes a two-fold levelling process. The bubble chamber is secured onto a bi-axial goniometer stage (Thorlabs, GNL20) which allows for easy positioning of the bubble after injection. This ensemble is mounted onto an aluminium base plate which is in turn attached to a

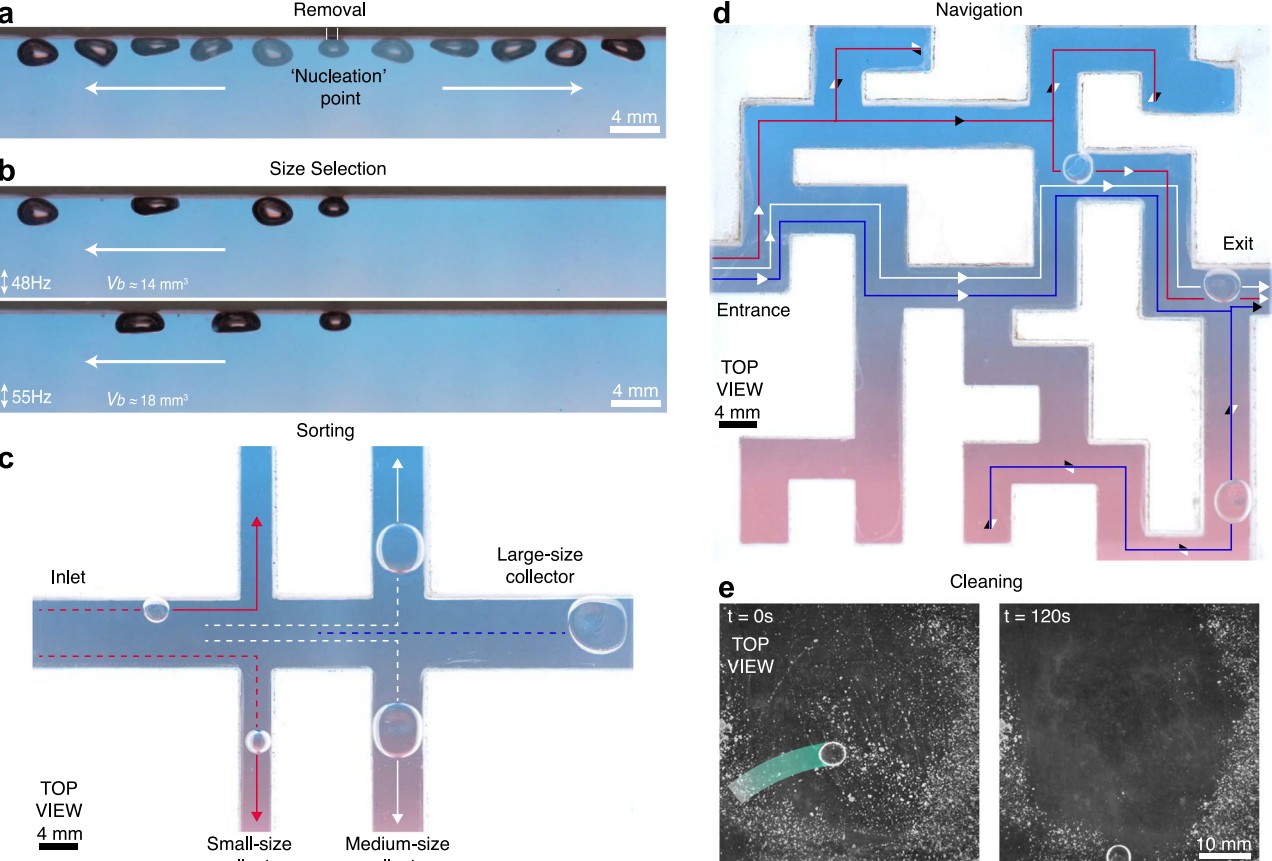

**Fig. 5 | Applications of galloping bubbles. a** Bubble evacuation: the galloping instability enables the removal of bubbles from a nucleation point, which hinder heat transfer in boiling. **b** Size selection: keeping the injection flow rate constant, bubble generation with tunable size becomes possible through the driving frequency, which determines when the bubbles start to gallop away from the nozzle. **c** Size-dependent sorting: owing to their affinity to follow lateral boundaries, bubbles of various volumes are autonomously directed into collectors of decreasing sizes, facilitating their sorting. **d** Navigation through complex networks: galloping bubbles have an ability to navigate intricate flow networks and solve mazes. The colored lines and arrows represent paths taken by different bubbles from the entrance until they reach the exit (red: $f = 50$ Hz at $A = 0.39$ mm, white: 45 Hz at 0.37 mm, and blue: 40 Hz at 0.54 mm). **e** Surface cleaning: particles covering a surface may be removed through the flows generated by bubbles exploring the domain randomly (See Supplementary Movie 5).

large circular plate equipped with three high-precision levelling screws. The levelling of the aluminium plate was carefully calibrated to ensure uniform acceleration across the entire base plate, as evidenced by a difference of less than 0.001 g between the readings of accelerations from the two acceleronmeters located in opposite ends of the plate. Before vibration is applied, the bubble is repositioned to the centre of the top wall using the goniometer, which is later re-levelled using precision micrometers.

Vibration – In our experiments, the liquid chamber is vibrated vertically by an electrodynamic shaker (Modal Shop, 2110E) and a power amplifier (Modal Shop, 2959E09) that produces a vertical displacement $z(t) = A \sin(\omega t)$, where $A$ and $f = \omega/2\pi$ are the maximum displacement and frequency, respectively. The shaker is connected to the base plate by a thin steel rod coupled with a linear air bearing (PI L.P. $4 \times 4''$ cross section, $6.5''$ long hollow bar) that ensures a spatially uniform vibration to within 0.1%[54]. The forcing is monitored using a data acquisition system (NI, USB-6243) with two piezoelectric accelerometers (PCB, 352C65), attached to the base plate on opposite sides of the drive shaft, and a closed-loop feedback ensures a constant acceleration amplitude to within $\pm 0.002$ g, where g is the gravitational acceleration[54].

Imaging – We image the shape evolution of vibrating bubbles from the side with a high-speed camera (Phantom, 410L) using a LED backlight (PHLOX LEDW BL) to generate high-contrast images of the bubble interface. For Fig. 1, a colour filter was placed between the backlight and the chamber to provide the colour gradient. To record the bubble's trajectory and galloping velocity, we tracked the bubble's position with a CCD camera positioned directly above the bubble chamber. The CCD camera was synchronized with the software driving the shaker to ensure that images were captured at the same point in every oscillation cycle, thus filtering out the bubble's oscillation and isolating their horizontal motion. For this configuration, the bubble was illuminated with a LED ring light, which allowed the bubble to be illuminated from all angles. The translational motion of the bubble was tracked with an in-house MATLAB algorithm. To visualise the flows generated by the oscillatory motion of the bubble interface, we use Particle-Image velocitmetry (PIV) by infusing the liquid with neutrally buoyant glass spheres (Sigma Aldrich, typical size 9–13 $\mu$m). A particle concentration of $\approx 1$ g/L was used. The mixture was shaken for at least 5 minutes to ensure the particles were sufficiently dispersed in the liquid before proceeding with the experiments. In the presence of a backlight, the tracer particles appear black in the video recordings and may thus be processed using a PIV code[55].

Galloping threshold–In experiments, the value of the galloping threshold, $A_G$, is sensitive to changes in room temperature and deviations in the levelling of the chamber's top surface. To minimize both of these effects, the top surface was re-levelled prior to each experiment (see 'Levelling'), and the experimental setup was allowed to vibrate for at least one hour before data collection. This allowed the

bubble chamber and surrounding environment to reach a steady temperature, which we monitored throughout the experiments.

## Simulations

We model galloping bubbles with the Navier-Stokes equations for two-phase flows in the single fluid formulation, which we solve with the open-source code Basilisk[56]. We regard the bubble as incompressible, since the maximum Mach number in the experiment is small, $Ma \approx 0.001 \ll 1$. We use the experimental fluid properties for the simulations; the liquid and gas densities are $\rho_l = 918$ kg m$^{-3}$, $\rho_g = 1$ kg m$^{-3}$ and the dynamic viscosities are $\mu_l = 4.59 \times 10^{-3}$ kg m$^{-1}$ s$^{-1}$, $\mu_g = 1.8 \times 10^{-5}$ kg m$^{-1}$ s$^{-1}$, respectively. The surface tension coefficient is $\sigma = 19.7$ mN m$^{-1}$. Both the liquid and gas are represented by a single fluid whose properties are the weighted average of the two phases by the volume fraction, $c(\mathbf{x}, t)$. Using the liquid properties as the characteristics scales, the dimensionless density and viscosity become $\rho(\mathbf{x}, t) = \rho_r c + (1 - c)$ and $\mu(\mathbf{x}, t) = \mu_r c + (1 - c)$, where $\rho_r$ and $\mu_r$ are the gas-to-liquid density and viscosity ratios, respectively. We obtain the dimensionless governing equations by using the characteristic time scale $\omega^{-1}$, flow velocity scale $A\omega$, and length scale $R$, which yield

$$\rho\left(\partial_t \mathbf{u} + \frac{A}{R}\mathbf{u} \cdot \nabla \mathbf{u}\right) = -\nabla p + \frac{A}{R}\frac{1}{Re}\nabla \cdot (2\mu\mathbf{D}) + \frac{R}{A}\frac{1}{We}\kappa\delta_s\mathbf{n} + \mathbf{G}(t),$$
$$\nabla \cdot \mathbf{u} = 0, \qquad (2)$$
$$\partial_t c + \frac{A}{R}\nabla \cdot (c\mathbf{u}) = 0,$$

where $\mathbf{u}(\mathbf{x}, t)$ and $p(\mathbf{x}, t)$ are the dimensionless velocity and pressure fields, and $Re = \rho_l A\omega R/\mu_l$ and $We = \rho_l \omega^2 R^3/\sigma$ are the Reynolds and Weber numbers, respectively. The viscous force is Newtonian with symmetric rate-of-strain tensor $\mathbf{D}$. The surface tension force acts at the fluid interface specified by the dimensionless Dirac delta distribution $\delta_s$ and the normal pointing from liquid to gas $\mathbf{n}$, which together are given by $\delta_s\mathbf{n} = \nabla c$. The curvature of the interface is $\kappa = \nabla \cdot \mathbf{n}$. The external sinusoidal forcing is incorporated as an effective gravity $\mathbf{G}(t) = \rho(-G + \sin t)\hat{\mathbf{z}}$ by working in the co-moving frame, where $G = g/(A\omega^2)$ is the dimensionless gravity and $\hat{\mathbf{z}}$ the unit vector in the vertical direction. For the hemispherical bubbles presented in this study, we choose $G = 0$ to remove deformations due to gravity in rest states, thus making them exactly hemispheres. Our simulations demonstrate that the galloping mechanism arises in microgravity and that bubbles in normal gravity exhibit similar dynamics.

The bubbles are simulated in a relatively large cubic domain of length $L = 16R$ to ensure that boundary effects from the bottom and lateral walls play no significant role. The top wall is modelled as a solid boundary with no-slip condition $\mathbf{u} = 0$. To simulate bubbles as those investigated in experiment for which the gas does not directly contact the wall, we impose a Dirichlet boundary condition on the volume fraction, $c = 0$. For the hemispherical bubbles that are attached to the wall, we use a Neumann boundary condition, $\partial_z c = 0$, which renders a freely moving contact line with 90 degree contact angle. Periodic boundary conditions are applied along the vertical boundaries, while a symmetric slip boundary condition is applied at the bottom boundary. Detached bubbles are initialized near the boundary to minimize transient effects.

The governing equations with these boundary conditions are solved on an octree discretization with adaptive mesh refinement applied at the interface and near large velocity gradients. We use a maximum resolution of 64 cells/$R$ to ensure the galloping speed is insensitive to refinement and that volume errors within a period $T$ are negligible. The bubble centroid is measured once the bubble has reached steady-state galloping, typically after $50T$. For each bubble volume, the galloping threshold, $A_G$, is defined as the midpoint between the driving amplitudes of the nearest stationary and galloping states. For the velocity collapse, we exclude Weber numbers that have

insufficient points to determine the threshold, near which resolving the precise value is difficult due to the slow onset of the instability. It is similarly numerically challenging to resolve galloping at large forcing amplitudes. Thus, we exclude bubbles that display nonphysical break-up or that have speeds that do not follow a monotonic increase from threshold. While the simulations capture the essence of the experimental observations, these numerical challenges and other effects lead to some differences between them. Owing to the very thin liquid film separating the bubble and solid wall, resolving the finer details of the intervening lubrication flow is particularly challenging. Similarly, the presence of surface roughness in the experimental setup, a factor not accounted for in simulations, may contribute to the simulated bubbles not displaying motion over a wider range of $We$ numbers relative to experiments. For the simulations shown in Supplementary Fig. S7, we seed massless tracer particles in the liquid phase and advect them with the flow field. We solve for the trajectories using a third-order Runge-Kutta integrator, with constraints to ensure the particles remain within the liquid phase.

## Theory

**Bubble spectrum.** Over a century ago, Lord Rayleigh[34] demonstrated that the shape oscillations of an incompressible fluid sphere (an immiscible drop or a gas bubble) immersed in an inviscid fluid are composed by a spectrum of Legendre polynomials $P_k(\cos\theta)$ with mode number $k \in \mathbb{N}$ that depend upon a polar angle $\theta \in [0, \pi]$. These vibration modes are axisymmetric about the $z$-axis, and oscillate with a natural frequency, $\omega_k$, given by the dispersion relation,

$$\omega_k^2 = (k^2 - 1)(k + 2)\frac{\sigma}{\rho R^3}. \qquad (3)$$

In a spherical bubble vibrating with frequency $\omega$, the mode $P_k(\cos\theta)$ will thus resonate when $\omega = \omega_k$. In dimensionless form, this condition defines a resonance Weber number, $We_k = \rho\omega_k^2 R^3/\sigma$, that turns the dimensional dispersion relation (3) into,

$$We_k = (k^2 - 1)(k + 2). \qquad (4)$$

Lamb[35] generalized Rayleigh's solution by demonstrating that the complete spectrum of a bubble includes an additional set of non-axisymmetric vibration modes with polar mode number, $k$, and azimuthal mode number $l \in [-k, k] \in \mathbb{Z}$, defined over an azimuthal angle $\varphi \in [0, 2\pi]$. These asymmetric modes oscillate at the same frequency (3) as their symmetric counterparts with mode number $k$. The complete set of vibration modes of a fluid sphere may thus be represented by the spherical harmonics, $Y_k^l(\theta, \varphi) = N_{kl}P_k^l(\cos\theta)e^{il\varphi}$, where $N_{kl}$ is a normalization constant. In the case of a hemispherical sessile bubble (or drop) with a free contact line, the vibration modes must satisfy the no-penetration condition at the wall. The spectrum is thus restricted to modes with $k + l =$ even, as the modes with $k + l =$ odd are not symmetric about the midplane, $z = 0$.

We consider a spherical system of coordinates $(r, \theta, \varphi)$ centered at the point where the bubble's centroid is projected onto the surface of the solid wall (Fig. 2g). At each instance in time, the axes are oriented such that the origin of the azimuthal angle aligns with the direction of the bubble motion. Relative to this moving system of coordinates, the interface position of a hemispherical bubble with static radius $R$ may thus be represented as $r(\theta, \varphi, t) = R + \eta(\theta, \varphi, t)$, where $\eta$ is the instantaneous interface deflection from the hemispherical shape. To characterize the deformations caused by the external forcing, we project the interface deflection onto the basis of spherical harmonics with $k + l =$ even, specifically $\eta(\theta, \varphi, t) = \sum_{k,l}c_{kl}(t)Y_k^l(\theta, \varphi)$. Using the standard approach, we leverage the orthogonality of the basis functions to

derive the instantaneous mode amplitudes, $c_{kl}(t)$, as follows,

$$c_{kl}(t) = \frac{\langle \eta, Y_k^l \rangle}{\langle Y_k^l, Y_k^l \rangle} = 2 \int_0^{2\pi} d\varphi \int_0^{\frac{\pi}{2}} d\theta (\sin \theta) \eta(\theta, \varphi, t) \overline{Y}_k^l(\theta, \varphi), \quad (5)$$

where $\overline{Y}_k^l$ denotes the complex conjugate of $Y_k^l$. We note that, for the hemispherical bubble, the vibration modes satisfy the following orthogonality conditions,

$$\langle Y_m^n, Y_k^l \rangle = \begin{cases} 1/2, & \text{if } m = k, n = l, \\ 0, & \text{otherwise} \end{cases} \quad (6)$$

with the standard normalization, $N_{kl}$, for spherical harmonics on the entire sphere. We then convert the resulting amplitudes $c_{kl}(t)$ for the complex-valued modes $Y_k^l(\theta, \varphi)$ into the amplitudes $a_{kl}(t)$ for the real-valued modes $Y_{kl}(\theta, \varphi)$, which we report in the main text.

The interface in our simulations is reconstructed numerically using piecewise linear facets. The facets in each numerical cell are computed using the volume fraction, $c$, and the normal, **n**, which together determine a unique cut between the phases of the cell. We use the centroids of the facets as the sampling of the interface position, $r(\theta, \varphi, t)$, which is later used to compute the deformation, $\eta(\theta, \varphi, t)$. To carry out the integration of (5), a discrete solid angle must be associated to each facet centroid. We obtain these by projecting the centroids onto the unit sphere, computing a spherical Voronoi diagram, and taking the Voronoi cell areas to be the solid angles[57].

An example of the instantaneous mode amplitudes for the three dominant modes in a galloping hemispherical bubble is shown in Supplementary Fig. S4. Since the amplitudes of these modes oscillate in time and depend on the driving amplitude, we use the $L_2$ norm, $\| a_{kl}(t)/A \|_2 = \sqrt{\int_0^T (a_{kl}(t)/A)^2 dt}$, to quantify their significance (Fig. 2h). In the parameter space explored in this work, *We* ~30–70, the first two axisymmetric modes, $Y_{20}$ and $Y_{40}$, have the highest amplitudes. When the interface oscillations become asymmetric and the bubble starts to gallop, the $Y_{31}$ harmonic is the most dominant non-axisymmetric mode. The remaining modes exhibit significantly smaller amplitudes, with their contribution diminishing as the mode order increases.

**Bubble speed.** In this section, we outline a mathematical method for calculating the steady speed of a galloping bubble expected from inertial forces to demonstrate that this self-propulsion mechanism represents a minimal realization of swimming in inviscid flow[23]. For a hemispherical bubble with static radius $R$, the problem is most favorably solved relative to a spherical system of coordinates $(r, \theta, \varphi)$ centered at the base of the bubble. The deformed liquid-gas interface may thus be defined as $r(\theta, \varphi, t) = R + \eta(\theta, \varphi, t)$. The flow velocity in the bulk is related to the interface deformation, $\eta(\theta, \varphi, t)$, by the kinematic condition,

$$\frac{\partial \eta}{\partial t} + \boldsymbol{u} \cdot \nabla \eta = \boldsymbol{u} \cdot \hat{\boldsymbol{r}}, \quad (7)$$

where, $\hat{\boldsymbol{r}}$ is the unit vector in the radial direction. The characteristic Reynolds number of galloping bubbles is $Re = A\omega R/\nu = 30$–$105$, which suggests that viscous effects are weak relative to inertial forces. In the incompressible, inviscid limit, $\nu = 0$, the flow velocity, $\boldsymbol{u} = \nabla \phi$, may be represented as the gradient of a scalar potential function, $\phi(r, \theta, \varphi, t)$, that satisfies Laplace's equation,

$$\nabla^2 \phi = 0. \quad (8)$$

The kinematic condition (7) thus relates the bubble deformation with the velocity potential at the interface, $\phi(R + \eta, \theta, \varphi, t)$.

Galloping bubbles demonstrate that interface deflections that are asymmetric over a period may cause a net translation. To obtain the bubble speed that the same interfacial motion would generate in a inviscid flow, we consider the total momentum balance in the horizontal direction $\hat{\boldsymbol{y}}$, assuming, without loss of generality, that the bubble gallops with velocity $v(t)$ in this direction. Since no external force is applied in any horizontal direction, when starting the system from rest, the momentum in the the galloping direction must always remain zero in the inviscid limit. The total horizontal momentum can be expressed as the sum of contributions from three sources: the momentum associated with the motion of the gas, the oscillatory flows relative to the bubble's center of mass, and the translation of the bubble[58]. We note that the momentum of the gas may be safely neglected compared to that of the liquid since the densities of the two phases differ by three orders of magnitude. The horizontal momentum generated by $\eta$ in the outer liquid, which has density $\rho$ and volume $V_l$, is given by $P_d = \rho \int_{V_l} (\boldsymbol{u} \cdot \hat{\boldsymbol{y}}) dV_l = \rho \int_S \phi(\hat{\boldsymbol{n}} \cdot \hat{\boldsymbol{y}}) dS$, where $\hat{\boldsymbol{n}}$ is the unit vector normal to the interface pointing into the gas. The conversion from an integral over the outer volume to an integral over the liquid-gas surface $S$ is enabled by the divergence theorem together with the flow decay in the far field, $\phi \to 0$ as $r \to \infty$. The momentum $P_d$ corresponds to the flow produced by the motion of the interface relative to the bubble's center of mass. Since the bubble's center of mass is translating, one must also consider the horizontal momentum associated with the displacement of the surrounding liquid, which may be expressed through the 'added mass', $M(t)$. We denoted the instantaneous velocity of the bubble's center of mass by $v(t)$, the horizontal momentum due to the added mass is thus $P_a = M(t)v(t)$. Therefore, the total momentum in the galloping direction of a bubble in inviscid flow is

$$P_a + P_d = 0, \quad (9)$$

which must be zero at all times if the system was initially at rest. In other words, if the integral of the shape-induced oscillatory flows is non-zero over a period, conservation of momentum (9) dictates that the body will undergo a net translation[58].

We use the momentum balance (9) to compute the steady velocity that the hemispherical galloping interface would generate in inviscid flow. To simplify the calculation, we consider the limit where the interface deformations are small relative to the bubble size, $\eta/R \ll 1$, which is appropriate near the bottom of the instability tongues where small forcing amplitudes lead to galloping ($A/R \lesssim 0.2$, Fig. 2F). For small deviations from the hemisphere, we may approximate the added mass as $M = \pi \rho R^3/3$, corresponding to half of the liquid mass that occupies the same volume as a rigid hemisphere undergoing translational motion. In this limit, the momentum balance (9) for a bubble galloping in the horizontal direction $\hat{\boldsymbol{y}}$ with velocity $\boldsymbol{v}(t) = v(t)\hat{\boldsymbol{y}}$ thus reduces to

$$\frac{\pi}{3} R^3 v(t) + \int_S \phi \hat{\boldsymbol{n}} \cdot \hat{\boldsymbol{y}} dS = 0. \quad (10)$$

The bubble speed expected from inertial forces, $v(t)$, may then be determined from the instantaneous interface deflection by solving the system of inviscid flow equations ((7), (8), (10)). To that end, we consider the general solution of (8) in terms of spherical harmonics, $\phi = \sum_{k,l} [b_{kl}(t)/r^{k+1}] Y_k^l(\theta, \varphi)$ for $r \geq R$, and express the interface deflection in the same basis, $\eta(\theta, \varphi, t) = \sum_{k,l} c_{kl}(t) Y_k^l(\theta, \varphi)$. Equations (7) and (10) require a knowledge of $\phi$ at the interface location, $r = R + \eta$, which may be mapped to the undeformed interface through a Taylor expansion as $\phi(r = R + \eta) \approx \phi(r = R) + \eta \partial\phi/\partial r(r = R)$; a suitable approximation in the small-deformation limit under consideration. For similar reasons, we may safely approximate $\hat{\boldsymbol{n}} \approx -\hat{\boldsymbol{r}}$ and $dS \approx (R + 2\eta) \sin \theta$. Given the interface deflection, $\eta$, we may thus use (7) to determine the potential amplitudes, $b_{kl}(t)$, and then (10) to determine the instantaneous speed, $v(t)$. By averaging the

instantaneous speed over an oscillation period, $T$, we may finally obtain the steady galloping speed, $V = \frac{1}{T}\int_0^T v(t)dt$ due to inertial forces.

We apply this inviscid theory to determine the galloping speed expected from inertial forces using the interfacial data from our simulations of hemispherical bubbles. We facilitate the numerical computation by truncating the bubble spectrum to the three dominant harmonics,

$$\eta \approx a_{20}(t)Y_{20}(\theta,\varphi) + a_{31}(t)Y_{31}(\theta,\varphi) + a_{40}(t)Y_{40}(\theta,\varphi), \qquad (11)$$

where $a_{kl}$ are the amplitudes of a real-valued harmonic basis (Section C1). Supplementary Fig. S4 shows the relative sizes of the amplitudes for a fixed set of parameters. The amplitudes of harmonics with $k > 4$ are increasingly smaller, and thus are not reported. The computed galloping speed, derived from this inviscid flow analysis, aligns well with the actual speeds observed in simulations (cf. Fig. 3d), indicating that the galloping self-propulsion can indeed be largely understood as swimming in an inviscid irrotational flow.

**Oscillator model.** The classical studies of Rayleigh[34] and Lamb[35] demonstrate that the amplitude of the vibration modes of a bubble in an inviscid flow (Section C1) follows the dynamics of a harmonic oscillator. In real fluids, these modes are subject to viscous effects and coupled by nonlinearities. Inspired by these studies, we present here a reduced oscillator model (Fig. 4a) that encapsulates the fundamental mechanism behind the self-propulsion of galloping bubbles. To simplify the problem, we concentrate the fluid mass, $M$, at a single point, which is subject to linear friction with damping constant $C$; an adequate approximation for small viscous effects in the fluid. To leading order, we approximate the restoring force provided by surface tension as a linear spring, with spring constant $k$ and equilibrium length $L$. Galloping bubbles are surrounded by an external liquid of density $\rho$ that subjects them to an inertial reaction force per unit volume that scales as $-\rho \mathbf{u}^2$, where $\mathbf{u}$ is the velocity field of the liquid. We incorporate this reaction force in our oscillator model through an external inertia acting on the point mass, specifically $\mathbf{F}_i = P|\dot{r}|\dot{r}$. Here, $P$ represents a coefficient akin to density, and $\dot{r}$ denotes the velocity of the point mass with $\mathbf{r}$ being the position vector relative to the pivot. The pendulum is subject to vertical vibrations with acceleration $a(t) = -A\omega^2\cos(\omega t)$, where $A$ is the maximum vertical displacement of the frame, and $\omega$ the vibration frequency. In the moving frame, the equation of motion for this pendulum thus becomes,

$$M\ddot{\mathbf{r}} + P|\dot{r}|\dot{r} + C\dot{\mathbf{r}} + k(r - L)\hat{\mathbf{r}} - Mg(t)\hat{\mathbf{x}} = 0, \qquad (12)$$

where $g(t) = g + a(t)$ is the effective gravity, $r = |\mathbf{r}|$ is the instantaneous pendulum's length, and $\hat{\mathbf{r}}$ and $\hat{\mathbf{x}}$ are the unit vectors in the radial and vertical directions, respectively.

By re-scaling length and time with $L$ and $\omega^{-1}$, respectively, we can rewrite (12) in terms of dimensionless variables as,

$$\ddot{\mathbf{r}} + p|\dot{r}|\dot{r} + \zeta\dot{\mathbf{r}} + \frac{1}{\kappa}(r - 1)\hat{\mathbf{r}} - (\delta + \varepsilon\cos t)\hat{\mathbf{x}} = 0, \qquad (13)$$

where $\kappa = M\omega^2/k$ denotes the pendulum's deformability, $\zeta = C/M\omega$ the friction coefficient, $p = PL/M$ the characteristic inertia of the surrounding medium, $\delta = g/L\omega^2$ the squared natural frequency, and $\varepsilon = A/L$ the dimensionless driving amplitude. Note that $\kappa$ tunes the pendulum stiffness; the spring becomes rigid in the limit when $\kappa = 0$, transforming our oscillator model into the well-known Kaptiza pendulum[49].

Stability—In analogy to the mechanism observed in galloping bubbles, we can demonstrate that in the deformable Kaptiza pendulum (13), the coupling between an oscillatory base state in the radial direction and an angular parametric instability results in a spontaneous symmetry breaking, ultimately leading the pendulum's self-propulsion. We begin by performing a stability analysis of the oscillator model using Floquet theory in the limit where the external inertia is weak, $p \ll 1$. In polar coordinates relative to the pivot, $(r, \theta)$, the components of (13) in the radial, $\hat{\mathbf{r}}$, and angular, $\hat{\boldsymbol{\theta}}$, directions thus become,

$$\ddot{r} - r\dot{\theta}^2 + \frac{(r - 1)}{\kappa} + \zeta\dot{r} - (\delta + \varepsilon\cos t)\cos\theta = 0, \qquad (14)$$

$$\ddot{\theta} + \left(\frac{2\dot{r}}{r} + \zeta\right)\dot{\theta} + \frac{1}{r}(\delta + \varepsilon\cos t)\sin\theta = 0. \qquad (15)$$

In the absence of driving, $\varepsilon = 0$, the pendulum's fixed points are $(r_0, \theta_0) = (1 + \delta\kappa, 0)$ and $(1 - \delta\kappa, \pi)$. When the pendulum is under vibration, $\varepsilon > 0$, we may look for a steady oscillatory solution about the fixed points of the form, $(r, \theta) = (r_0 + R_0(t), 0)$. By substituting in (14), we find that $R_0(t)$ evolves as a forced damped oscillator,

$$\ddot{R}_0 + \zeta\dot{R}_0 + \frac{1}{\kappa}R_0 = \varepsilon\cos t, \qquad (16)$$

which has a particular solution, $R_0(t) = \varepsilon\kappa[(\kappa - 1)\cos t + \zeta\kappa\sin t]/[(1 - \kappa)^2 + (\zeta\kappa)^2]$. For a deformable pendulum, $\kappa > 0$, this solution corresponds to an oscillatory base state where the point mass moves up and down about the fixed points (Fig. 4c). Focusing on the lower fixed point, $r_0 = 1 + \delta\kappa$, we investigate the stability of the oscillatory base state by assuming a small perturbation on the steady solution,

$$r(t) = r_0 + R_0(t) + R(t), \qquad (17)$$

$$\theta(t) = 0 + \varphi(t), \qquad (18)$$

where $R \ll 1$ and $\varphi \ll 1$. To a first-order approximation, the governing equations ((14),(15)) become,

$$\ddot{R} + \zeta\dot{R} + \frac{1}{\kappa}R = 0, \qquad (19)$$

$$\ddot{\varphi} + \left(\frac{2\dot{R}_0}{r_0 + R_0} + \zeta\right)\dot{\varphi} + \left(\frac{\delta + \varepsilon\cos t}{r_0 + R_0}\right)\varphi = 0. \qquad (20)$$

Perturbations in the radial direction are thus exponentially suppressed according to an unforced damped oscillator equation (19). However, angular perturbations are governed by parametrically forced equation (20), which includes significantly richer dynamics. We note that in the rigid-pendulum limit, $\kappa = 0$, the angular equation (20) reduces to the classical damped Mathieu's equation, whose stability is well-known[59]. As the pendulum becomes deformable, $\kappa > 0$, angular perturbations evolve according to a generalized Mathieu's equation with additional periodic coefficients due to the stable base state, $R_0(t)$, which oscillates with the same frequency as the external forcing. To analyze the angular stability of the deformable pendulum, we thus write (20) as a first-order system,

$$\dot{\mathbf{u}} = \begin{bmatrix} 0 & 1 \\ -\dfrac{\delta + \varepsilon\cos t}{r_0 + R_0} & -\left(\dfrac{2\dot{R}_0}{r_0 + R_0} + \zeta\right) \end{bmatrix}\mathbf{u}, \qquad (21)$$

where $\mathbf{u} = [\varphi, \Omega]^T$ and $\Omega = \dot{\varphi}$. Floquet theory indicates that the solutions to (21) are of the form $\mathbf{u}_i(t) = e^{\mu_i t}\mathbf{p}_i(t)$, where $\mu_i$ is the characteristic Floquet exponent, and $\mathbf{p}_i(t)$ a periodic function with the same period, $T = 2\pi$, as the coefficient matrix. We define a fundamental solution

matrix to this system as,

$$X(t) = \begin{bmatrix} \phi_1(t) & \phi_2(t) \\ \Omega_1(t) & \Omega_2(t) \end{bmatrix}, \tag{22}$$

where the columns are the solutions with the following initial conditions,

$$X(0) = \begin{bmatrix} 1 & 0 \\ 0 & 1 \end{bmatrix}. \tag{23}$$

By numerically integrating (21) over one period, $T$, from the initial conditions in (23), we may compute the characteristic matrix of the problem,

$$B = \begin{bmatrix} \phi_1(T) & \phi_2(T) \\ \Omega_1(T) & \Omega_2(T) \end{bmatrix}. \tag{24}$$

The stability of the system is determined by the eigenvalues, $\lambda_i$, of the matrix $B$. If $|\lambda_i| > 1$ for either of the two eigenvalues, the system is unstable, exhibiting exponential growth in time. If $|\lambda_i| < 1$ for the two eigenvalues, the system is stable and the solutions are quasi-periodic. This approach may thus be used to determine the neutral stability curves, or 'tongues', that separate the stable and unstable regions in the frequency-forcing phase map. Supplementary Fig. S5a illustrates the harmonic and subharmonic tongues in the absence of damping, $\zeta = 0$, for increasing pendulum deformability. When the spring is rigid, $\kappa = 0$, the neutral curves correspond to those of the classic Mathieu equation, where the tongues intersect the $\varepsilon$-axis at the resonant frequencies $\delta = n^2/4$, with $n$ a positive integer. Note that $n = 1$ and $n = 2$ correspond to the first subharmonic and harmonic instabilities, respectively. As the spring's deformability increases, $\kappa > 0$, the resonances shift toward higher frequencies. When the system is subject to friction, $\zeta > 0$, the tongues no longer intersect the $\varepsilon$-axis; the driving must exceed a critical threshold for the instability to arise at all frequencies, $\delta$ (Supplementary Fig. S5b). This instability condition is equivalent to the galloping threshold, $A_G$, reported in the main text for vibrating bubbles.

Thus, when the pendulum (13) is vibrated with an amplitude $A > A_G$, which depends on the frequency, small lateral perturbations grow in the angular direction, $\hat{\boldsymbol{\theta}}$, superimposed on a pre-existing oscillatory base state in the radial direction, $\hat{\boldsymbol{r}}$, given by (16). These parametrically-excited azimuthal oscillations exhibit exponential growth over time and are eventually saturated by the system's nonlinearities. Once equilibrium is reached, the pendulum's motion results from a combination of vertical and lateral oscillations, causing the mass to follow a curved trajectory and acquire angular momentum (Fig. 4d). Similar to galloping bubbles, the coupling between a stable oscillatory base state and a parametric instability is thus the essential mechanism behind the spontaneous symmetry breaking of our deformable pendulum.

Galloping pendulum—Owing to the chiral motion of the mass arising when $A > A_G$, the pivot experiences a force whose horizontal component is $F_h = k(r - L)\sin\theta$ in dimensional form. If the pivot is allowed to slide along a horizontal rail (Fig. 4d), this force may generate a net translation of the pendulum. To demonstrate such self-propulsion of our vibrating pendulum, we thus consider the generalized problem that includes a sliding pivot with point mass $M_p$, and position $\boldsymbol{r}_p = (0, y_p)$ relative to a system of coordinates moving with the frame. We assume that the pivot is subject to friction with a linear damping coefficient $D$. The position of the hanging mass relative to the same system of coordinates is $\boldsymbol{r}_m = \boldsymbol{r}_p + \boldsymbol{r}$, where $\boldsymbol{r}$ remains the position of the hanging mass relative to the pivot. The force balance on the hanging mass is now coupled with the force balance on the pivot. After non-dimensionalizing the system using the same characteristic scales

as in (13), the equations of motion for the sliding-oscillator system become,

$$\xi \ddot{y}_p + \zeta_p \dot{y}_p - \tfrac{1}{\kappa}(r - 1)\sin\theta = 0,$$
$$\ddot{\boldsymbol{r}}_m + p|\dot{\boldsymbol{r}}_m|\dot{\boldsymbol{r}}_m + \zeta \dot{\boldsymbol{r}}_m + \tfrac{1}{\kappa}(r-1)\hat{\boldsymbol{r}} - (\delta + \varepsilon\cos t)\hat{\boldsymbol{x}} = 0. \tag{25}$$

Here, $r = |\boldsymbol{r}|$, $\sin\theta = y_m/r$, and the new dimensionless numbers are $\xi = M_p/M$ and $\zeta_p = D/M\omega$, representing the mass ratio and the pivot friction coefficient, respectively. Combining the horizontal force balances in (25) yields the equation of motion for the horizontal position of the center of mass of the system, denoted by $y_{cm}$,

$$(1 + \xi)\ddot{y}_{cm} = -p|\dot{\boldsymbol{r}}_m|\dot{\boldsymbol{r}}_m \cdot \hat{\boldsymbol{y}} - \zeta_p \dot{y}_p - \zeta \dot{\boldsymbol{r}}_m \cdot \hat{\boldsymbol{y}}. \tag{26}$$

These equations may be solved numerically to obtain the instantaneous horizontal speed of the pendulum, $\dot{y}_{cm}(t)$, whose average over an oscillation period, $V_{cm} = (1/T)\int_0^T \dot{y}_{cm}dt$, yields the steady self-propulsion speed. In the absence of the nonlinear term, the forcing from the friction terms in (26) average to zero. We thus note that the inertial term, $p|\dot{\boldsymbol{r}}_m|\dot{\boldsymbol{r}}_m$, is a key ingredient for generating steady net motion, providing a non-zero forcing over the oscillation period that leads to propulsion.

In Fig. 4d, e, we illustrate the steady self-propulsion predicted by the oscillator model through numerical solutions of equations (25). Parameters for the sliding pivot were chosen as $\zeta_p = 0.2$, $\xi = 100$, so that the trajectory of the hanging mass is not significantly influenced by the comparatively smaller displacement of the pivot. The steady propulsive speed of the pivot, $V_{cm}$, exhibits a direct dependence on the driving amplitude beyond the self-propulsion threshold, similar to what occurs for galloping bubbles (cf. Fig. 3c). The reduced oscillator model may thus be considered as a minimal realization of the galloping mechanism.

## Data availability
The data supporting the findings in this study are available within the paper and its Supplementary Information. Source data are provided with this paper.

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

## Acknowledgements

The authors thank Austin M. Blitstein for helpful discussions and Xinyun Liu for collecting additional experimental data. The authors gratefully acknowledge financial support from the Alfred P. Sloan Foundation through a Sloan Research Fellowship (P.J.S.) and the U.S. National Science Foundation through CMMI-2321357 (P.J.S., J.H.G.) and CAREER award CBET-2144180 (P.J.S.). The authors thank the Research Computing group at the University of North Carolina at Chapel Hill for providing computational resources.

## Author contributions

J.H.G. performed the experiments. C.W.M. conducted the bubble simulations. H.A.S. and P.J.S. discussed simplified models. S.I.T. and P.J.S. performed the theoretical analysis. P.J.S. directed the project. J.H.G., S.I.T., C.W.M., H.A.S., and P.J.S. discussed the results, including

experiments and modeling, and contributed to writing the paper and revising.

## Competing interests

The authors declare no competing interests.
