## [Transparent Peer Review file · Nature Communications]

Galloping Bubbles

Corresponding Author: Professor Pedro Sáenz

Version 0:

Reviewer comments:

Reviewer #1

(Remarks to the Author)

The enclosed manuscript describes spontaneous symmetry breaking in air bubbles driven by periodic gravitational oscillations, which leads to steady propulsion. Several interesting observations are made including various modes of motion. Moreover, the authors do a fantastic job of looking at a simple system and modeling it in such a way that it can be fully understood. The introduction of several applications is interesting, but leaves more to be desired. This will certainly be of interest to the community and recommend publication in this journal after they address the following remarks:

- Scale bars in figures and videos would be helpful to orient the viewer.
- I'm confused by the term "breakup" on page 2 and figure 1e. Does this mean the bubbles collapse and disappear? Break down into smaller pieces?
- Authors introduce V_{bath} at the end of page 2 but never define it. Additionally, in some communities, speed in body lengths per second would be a useful metric for comparison.
- If I understand correctly, the speed and behavior depends only on amplitude. Please mention if the influence of driving frequency has been explored in this system.
- The size and shape of the markers in figure 2e are not defined, and not intuitive enough to decipher. Do the colors in the boxes directly correspond to the speed color bar at the bottom? Please describe their meaning.
- The authors pose the question, "Is the galloping symmetry breaking exclusive to bubbles, or can this propulsion mechanism be realized in other systems?" and end with an explanation based on K. Is a bubble the only material that meets those criteria?
- I find the Proof of Concept Section significantly lacking, and deserving of more attention. A lot of cool stuff here. The theory is well-developed, so what is the origin of this attraction to vertical walls (symmetry arguments presumably)? How does that lead to navigation of mazes or sorting? Even if the answer is obvious, please explain to your audience.
- o To this end, what does the coloration in figures 5C and 5D mean. Is this just a part of the experimental setup or was it added artificially to represent something? If not necessary, simply make the images black and white as not to cause confusion.
- Even in the case of cleaning, is the bubble simply pushing particles around, or is there more at play, such as interfacial capillary forces? This would be very useful to know.
- In this system, it is clear that the mechanism is oscillation due to gravity, but is the system open and at constant pressure, or closed and also undergoing pressure oscillations? Could this model be generalized to bubbles undergoing periodic pressure oscillation (i.e. in an acoustic field, which is an active area of research)? A strict comment isn't necessary, but it might be interesting for the authors to consider.

Reviewer #2

(Remarks to the Author)

This paper reports on a remarkable and potentially useful phenomenon, the spontaneous motion of bubbles in a container subject to vibrations which are sufficiently strong. Some possible applications are illustrated convincingly in proof-of-concept experiments.

The main features of the spontaneous bubble motion are mapped out in experiments: the symmetry breaking instability leading to motion on the one hand, and a map of different modes of motion on the other

hand.

As for a theoretical understanding of the motion, a variety of different approaches are used, and explained in more detail in the supplementary information.

First, a full numerical simulation of the Navier-Stokes equation, yielding good agreement with experiment. Second, a mode analysis of the motion, based on Rayleigh's theory of oscillating bubbles. Third, a reduced mass-spring model, which in large parts is amenable to analytical analysis.

My only minor concerns are that in the main text, the oscillator model does not seem to include propulsion, which seems to be the main feature of the phenomenon. It would therefore be useful to at least indicate in the main text how (1) couples to horizontal motion.

In the "Bubble speed" section of the SI, I was wondering how in an inviscid theory a finite bubble speed is reached. Presumably there must be some dissipative mechanism to keep the bubble from speeding up forever.

In conclusion, I highly recommend this paper for publication.

Version 1:

Reviewer comments:

Reviewer #1

(Remarks to the Author)

The authors have done a great job of responding to my comments. I recommend their manuscript for publication.

Reviewer #2

(Remarks to the Author)

The authors have been very conscientious with their responses to both referees. The changes have improved further the readability of the manuscript.

I recommend publication in the present form.

Response to Reviewer 1

J. H. Guan, S. I. Tamim, C. W. Magoon, H. A. Stone, P. J. Sáenz
Department of Mathematics, University of North Carolina
Department of Mechanical & Aerospace Engineering, Princeton University

(Dated: November 23, 2024)

We thank the referee for their careful reading of our manuscript, and their constructive review. We respond below to the points raised in their report. In what follows, the referee’s comments are reproduced in *black italics*. We have taken the referee’s comments on board and revised the manuscript accordingly.

All changes in the main text and appendices have been highlighted in blue.

The enclosed manuscript describes spontaneous symmetry breaking in air bubbles driven by periodic gravitational oscillations, which leads to steady propulsion. Several interesting observations are made including various modes of motion. Moreover, the authors do a fantastic job of looking at a simple system and modeling it in such a way that it can be fully understood. The introduction of several applications is interesting, but leaves more to be desired. This will certainly be of interest to the community and recommend publication in this journal after they address the following remarks:

We were delighted to read the referee’s appreciation of our work, and hope that our responses to their questions meet their expectations, leading to a recommendation for publication.

- Scale bars in figures and videos would be helpful to orient the viewer.

Agreed. Scale bars have been added to the supplementary videos. We have also verified that all figures in the manuscript include scale bars.

- I’m confused by the term “breakup” on page 2 and figure 1e. Does this mean the bubbles collapse and disappear? Break down into smaller pieces?

Our experiments are conducted in a relatively mild flow regime (the maximum Mach number $Ma \approx 0.001 \ll 1$, as reported in the SI), where the gas within the bubbles remains incompressible. Bubble collapse as that observed during cavitation does not occur in our galloping regime. By “breakup” we thus refer to the process by which the bubble splits into multiple smaller bubbles, with the total volume being conserved. This breakup can happen in two ways: i) the original bubble splitting into two similarly size bubbles when the amplitude of the ‘dimple’ becomes comparable to the bubble radius, or ii) the ejection of a smaller bubble when the oscillatory modes create elongated interfaces that eventually pinch off.

We have added a remark in the main text to clarify this point.

- Authors introduce V_{bath} at the end of page 2 but never define it. Additionally, in some communities, speed in body lengths per second would be a useful metric for comparison.

Although we had defined the average bath speed, $V_{bath} = 4A/T$, we agree that reordering the sentence could improve clarity. We have made this adjustment in the revised text.

We appreciate the reviewer’s insightful suggestion that reporting the galloping speed in body lengths per unit time could help connect and contextualize our finding within the broader swimming literature. Using the bubble radius, R , as the characteristic body length and the driving frequency, f , as the natural beat frequency, we find that galloping bubbles swim with speeds as high as $V \sim 0.5Rf$, in the typical range for fish and cetaceans (Sánchez-Rodríguez *et al. Nat. Comms* 2023). We have added a sentence in the main text to highlight this observation.

- If I understand correctly, the speed and behavior depends only on amplitude. Please mention if the influence of driving frequency has been explored in this system.

Once a bubble enters the galloping regime, its speed becomes strongly dependent on the amplitude beyond the galloping threshold, as illustrated in Fig. 3c. However, we recall that a prerequisite for galloping is that the bubble’s Weber number must be near $We = \rho\omega^2 R^3/\sigma \sim 40$. For a given fluid, this condition can be met by varying either the driving frequency, ω , or the bubble size, R . We chose the latter to better illustrate and support our hemispherical approximation. The influence of frequency is thus illustrated in Fig. 2e, where the galloping phase map is presented in dimensionless terms of We . Additionally, we have updated Fig. 3c with an improved scaling for the galloping speed which includes the We number and shows a better collapse of the experimental and numerical data.

- The size and shape of the markers in figure 2e are not defined, and not intuitive enough to decipher. Do the colors in the boxes directly correspond to the speed color bar at the bottom? Please describe their meaning.

All markers are square, with both their size and color representing the galloping speed observed in our simulations. The color bar at the bottom quantifies this speed. This representation style was chosen to be able to include simulations that did not result in motion without cluttering the phase map. We have now clarified the meaning of the marker sizes in the caption for Fig. 2.

- The authors pose the question, “Is the galloping symmetry breaking exclusive to bubbles, or can this propulsion mechanism be realized in other systems?” and end with an explanation based on K . Is a bubble the only material that meets those criteria?

Our reduced-order oscillator model demonstrates that the galloping locomotion relies on rather generic ingredients, suggesting that this form of symmetry breaking could indeed be observed in other systems. Within this reduced framework, the bubble is depicted as a deformable body, with the spring force (characterized by the material constant k) serving as a minimal model for a restoring force (originating from surface tension in the case of bubbles). Systems with analogous elements could be identified with relative ease. For instance, elasticity provides a natural restoring force in solid materials. Our reduced model thus indicates that by leveraging the elasticity of a deformable shell in a vertically vibrated inertial fluid, it may be possible to generate propulsion, as this system contains all the necessary elements for the galloping symmetry breaking. This thought experiment also suggests that our study could inspire new research in soft-robotic swimmers.

- I find the Proof of Concept Section significantly lacking, and deserving of more attention. A lot of cool stuff here. The theory is well-developed, so what is the origin of this attraction to vertical walls (symmetry arguments presumably)? How does that lead to navigation of mazes or sorting? Even if the answer is obvious, please explain to your audience.

We have performed additional numerical simulations to illustrate the fluid-mediated attraction of bubbles to lateral walls in more detail (Supplementary Fig. 6). The vertical cross-section of the oscillating bubble reveals an additional symmetry-breaking of the interface as well as the external flows; the stroke pushing the bubble into the wall is stronger than the one away from it, which results in a net force that holds the bubble against the wall. This inherent attraction to lateral walls underpins the bubble’s self-sorting and maze-navigation capabilities illustrated in Fig. 5. For bubbles following walls, a straightforward sorting mechanism arises when outlets are arranged in increasing size. In navigating mazes, bubbles exemplify the simple yet effective “wall-follower” algorithm, or “right-hand rule”, which states that a maze without islands can be solved by simply following a lateral wall.

We have added a remark on page 5 of the main text, and included a new figure (Supplementary Fig. 6)

to illustrate the attraction of galloping bubbles to walls.

o To this end, what does the coloration in figures 5C and 5D mean. Is this just a part of the experimental setup or was it added artificially to represent something? If not necessary, simply make the images black and white as not to cause confusion.

The color gradient is used purely to enhance the visual appeal and engage the broad audience of *Nature Communications*. We have added a remark in the caption of the first figure to clarify this point throughout the paper.

- Even in the case of cleaning, is the bubble simply pushing particles around, or is there more at play, such as interfacial capillary forces? This would be very useful to know.

To illustrate how a surface cleaning task is performed by galloping bubbles, we have performed numerical simulations incorporating tracer particles. The results are illustrated with an accompanying figure (Supplementary Fig. 7). Our simulations show that particles near the wall are advected downward by the oscillation-induced flows (Supplementary Fig. 7b).

Moreover, we have performed additional experiments to verify that the particles removal from the surface is primarily due to the galloping flows, rather than gravity or the vertical oscillations. In our cleaning experiments, we used glass spheres (Sigma Aldrich, radii 9-13 μm), which naturally adhere to the top wall. To confirm that the removal was not driven by gravity (i.e., particles simply sinking), we spread particles on the underside of the top wall, filled the chamber with liquid and allowed it to rest for 12 hours. We found no noticeable change in the number of particles on the wall. To ensure that vertical vibrations alone were not responsible for particle removal either, we repeated the test with the fluid chamber subjected to vertical oscillations under similar driving conditions to those used in Fig. 5e. Once again, we observed no significant depletion of particles from the top surface. We have included a remark on page 6 to indicate the results of these experimental tests.

- In this system, it is clear that the mechanism is oscillation due to gravity, but is the system open and at constant pressure, or closed and also undergoing pressure oscillations? Could this model be generalized to bubbles undergoing periodic pressure oscillation (i.e. in an acoustic field, which is an active area of research)? A strict comment isn't necessary, but it might be interesting for the authors to consider.

We designed the fluid chamber with an opening to ensure that the reference pressure is atmospheric. We thank the referee for noting this omission, which we have now addressed in SI Section A. Experiments in the revised text.

Indeed, the potential to achieve galloping locomotion through acoustic forcing presents an exciting opportunity, especially given the extensive research in this area. Although the existing literature on acoustically driven bubbles does not report steady propulsion similar to that of galloping bubbles, we see no fundamental reason why sessile bubbles subjected to an acoustic field could not exhibit the vibration modes responsible for galloping. A particularly interesting aspect to investigate will be the influence of volume oscillations, which are characteristic of the acoustic problem but absent in our system.

We thank the referee once again for their helpful and constructive review of our work, and hope that our responses and revisions meet their expectations.

Response to Reviewer 2

J. H. Guan, S. I. Tamim, C. W. Magoon, H. A. Stone, P. J. Sáenz
Department of Mathematics, University of North Carolina
Department of Mechanical & Aerospace Engineering, Princeton University

(Dated: November 23, 2024)

We thank the referee for their careful reading of our manuscript, and their constructive review. We respond below to the points raised in their report. In what follows, the referee’s comments are reproduced in *black italics*. We have taken the referee’s comments on board and revised the manuscript accordingly.

All changes in the main text and appendices have been highlighted in blue.

This paper reports on a remarkable and potentially useful phenomenon, the spontaneous motion of bubbles in a container subject to vibrations which are sufficiently strong. Some possible applications are illustrated convincingly in proof-of-concept experiments.

The main features of the spontaneous bubble motion are mapped out in experiments: the symmetry breaking instability leading to motion on the one hand, and a map of different modes of motion on the other hand.

As for a theoretical understanding of the motion, a variety of different approaches are used, and explained in more detail in the supplementary information. First, a full numerical simulation of the Navier-Stokes equation, yielding good agreement with experiment. Second, a mode analysis of the motion, based on Rayleigh’s theory of oscillating bubbles. Third, a reduced mass-spring model, which in large parts is amenable to analytical analysis.

We are delighted to read the referee’s appreciation of our work.

My only minor concerns are that in the main text, the oscillator model does not seem to include propulsion, which seems to be the main feature of the phenomenon. It would therefore be useful to at least indicate in the main text how (1) couples to horizontal motion.

While the mechanism by which the pendulum coupled to a sliding pivot generates steady motion was detailed in the Supplementary Information, we agree that the main text could benefit from a more detailed discussion. We have therefore added the following sentences, which includes a specific reference to the coupled system (25), on page 5 of the revised manuscript:

“To illustrate how this symmetry breaking may lead to self-propulsion, we consider the pendulum to be attached to a sliding pivot with position $(0, y_p(t))$, mass M_p , and frictional coefficient D (Fig. 4, and SI, Theory). By solving the resulting coupled system (25, SI) in a regime where the pivot displacement is small relative to that of the pendulum (e.g. $M_p \gg M$), the motion of the swinging mass remains largely unaffected by the pivot’s motion and generates a net horizontal force that propels the pendulum forward (Fig. 4d) at a steady speed, which increases with the distance beyond the instability threshold (Fig. 4e).”

In the “Bubble speed” section of the SI, I was wondering how in an inviscid theory a finite bubble speed is reached. Presumably there must be some dissipative mechanism to keep the bubble from speeding up forever.

We note that no external force is applied in the direction along which the bubble gallops. Therefore, when starting the system from rest, the momentum in the horizontal direction must always remain zero. Denoting $v(t)$ as the instantaneous velocity of the bubble’s center of mass in the same direction, the total

horizontal momentum can be expressed as the sum of contributions from three sources: the momentum associated with the bubble’s mass, $mv(t)$; the fluid momentum due to the instantaneous translation, $M(t)v(t)$ (where $M(t)$ represents the ‘apparent’ or ‘added’ mass); and the fluid momentum arising from the interface deformations relative to the bubble’s center of volume, $P_D(t) = -\rho \int_S \phi \hat{\mathbf{n}} \cdot \hat{\mathbf{y}} dS$. Hence, horizontal momentum conservation means,

$$(m + M(t)) v(t) - \rho \int_S \phi \hat{\mathbf{n}} \cdot \hat{\mathbf{y}} dS = 0, \quad \rightarrow \quad v(t) = \frac{\rho \int_S \phi \hat{\mathbf{n}} \cdot \hat{\mathbf{y}} dS}{m + M(t)}.$$

In other words, given periodic body deformations, if the integral of the oscillatory flows over a period is non-zero, conservation of momentum dictates that the body must experience a net translation (see section 4.4. in Childress, S. *An Introduction to Theoretical Fluid Mechanics*. Vol. 19. AMS, 2009.). We note that this result is not in contradiction to D’Alembert’s paradox: one can demonstrate that the average force over a period on the body (and equivalently on the fluid) is always zero, given that the flow is inviscid and irrotational (see Childress 2009). From this momentum balance, it readily follows that achieving a finite speed is contingent upon achieving finite-amplitude oscillations. In fluid systems like ours, saturation mechanisms that lead to such finite-amplitude oscillations typically arise from nonlinearities, including those associated with the system’s restoring force, which in our case is surface tension. Hence, dissipation is not strictly necessary to achieve a finite speed. However, we find that viscosity helps expand the parameter space for coherent motion by reducing the number of mixed modes, as the presence of too many often leads to chaotic dynamics that prevent coherent motion. We have added some clarifying remarks to improve the discussion in the SI.

We thank the referee once again for their helpful and constructive review of our work, and hope that our responses and revisions meet their expectations.